# Open Data Resources on COVID-19 in Six European Countries: Issues and Opportunities

**DOI:** 10.3390/ijerph181910496

**Published:** 2021-10-06

**Authors:** Fabrizio Pecoraro, Daniela Luzi

**Affiliations:** Institute for Research on Population and Social Policies, National Research Council, Via Palestro, 32, 00185 Rome, Italy; d.luzi@irpps.cnr.it

**Keywords:** open datasets, COVID-19, data quality, data reusability, epidemiological data

## Abstract

Since the beginning of the COVID-19 pandemic in March 2020, national and international authorities started to develop and update datasets to provide data to researchers, journalists and health care providers as well as public opinion. These data became one of the most important sources of information, which are updated daily and analysed by scientists in order to investigate and predict the spread of this epidemic. Despite this positive reaction from both national and international authorities in providing aggregated information on the diffusion of COVID-19, different challenges have been underlined in previously published studies. Different papers have discussed strengths and weaknesses of these types of datasets by focusing on different quality perspectives, which include the statistical methods adopted to analyse them; the lack of standards and models in the adoption of data for their management and distribution; and the analysis of different data quality characteristics. These studies have analysed datasets at the general level or by focusing the attention on specific indicators such as the number of cases or deaths. This paper further investigates issues and opportunities in the diffusion of these datasets under two main perspectives. At the general level, it analyses how data are organized and distributed to scientific and non-scientific communities. Moreover, it further explores the indicators adopted to describe the spread of the COVID-19 epidemic while also highlighting the level of detail used to describe them in terms of gender, age ranges and territorial units. The paper focuses on six European countries: Belgium, France, Germany, Italy, Spain and UK.

## 1. Introduction

The number of individuals infected with severe acute respiratory syndrome coronavirus 2 (SARS-CoV-2), the virus causing coronavirus disease 2019 (COVID-19), is still spreading dramatically across the globe since its outbreak in China [1] and was declared a pandemic by the WHO (World Health Organization) on 11 March 2020. In Europe, the first tested and declared positive case for SARS-CoV-2 was managed by the hospital of Codogno, Italy, on 20 February 2020. From this case, a rapidly increasing number of patients have been identified, initially in Northern Italy and later in the rest of the country and Europe. Accessible epidemiological data are of great value for emergency preparedness and response for understanding disease progression and building statistical and mechanistic disease models that enable forecasting and can be used in decision making and the allocation of resources [2].

The value of accessible sources of information for emergency preparedness and the need for better data sharing have already been confirmed during the first stages of recent outbreaks such as Ebola and Zika [3]. The prompt distribution of high-quality data can positively influence the vaccination uptake [4]. Moreover, it provides crucial information to policy makers and hospitals professionals for anticipating surge capacity and resources allocation during a massive outbreak, such as beds, professionals and devices [5,6].

The availability of open data is even more important considering the widespread distribution of complex and effective statistical predictive models [7] that may help in identifying the timing of the maximum prevalence of different pandemic waves along with its amplitude and duration [8], thus allowing the prediction and forecast of the epidemic spread across regions and countries [9,10,11] at the beginning of the pandemic.

The provision of rapid, accurate and open access data became one of the first non-clinical challenges emphasized by the COVID-19 epidemic [12] aimed at providing detailed information for different purposes and to various professionals such as researchers, journalists, health care providers and public opinion [13,14,15]. The distribution of these datasets to keep the public informed and to support policymakers in refining interventions [16] indicates the extremely high importance in allowing open access to data in order to develop potentially useful models with high-precision results. Moreover, these datasets have been adopted to define and update international sources of information, such as the WHO, European Centre for Disease Prevention and Control (ECDC) and Johns Hopkins University (JHU), that are widely adopted to monitor trends in the virus outbreak and to assess the risks of the pandemic in several countries and regions all over the world [17]. However, different challenges have been underlined in previous studies published during these months. One of the first studies within this context has been published during the beginning of the pandemic in Europe by Alamo et al. [18], providing a review of the main open resources for addressing the COVID-19 pandemic from a data science point of view. In this study, the authors underlined the many opportunities provided by the availability of open-data resources on COVID-19 and related variables. However, they also pointed out that these analyses may be corrupted by data inconsistency, changing criteria, a large diversity of sources, non-comparable metrics between countries, delays, etc. The widespread distribution of comparable epidemiological data is complicated considering that, while data standards exist for observational studies and clinical research, no such standards have been developed yet for public health-related epidemiological data [12,19]. This unavailability of a common data model as well as of vocabularies to represent data elements make global comparisons a challenging task [16]. Other common challenges have been underlined by Batker and colleagues [20], posing the attention on the heterogeneity of information sources, the quality of data and metadata and the comparability and reusability of information gathered from multiple sources. Similarly, in a parallel study [21] we investigated the compliance of datasets against the 15 FAIR (Find, Access, Interoperate and Reuse) principles [22] as well as their quality characteristics. The main findings of this study highlight the absence of interoperability and reusability of both data and metadata, while from a quality perspective, institutional datasets should be improved in particular in terms of traceability and understandability. Both characteristics are affected by the lack of detailed information in terms of definition and formulas adopted to compute each indicator as well as data flow describing the collection, elaboration, aggregation and distribution of data. Other studies (e.g., [23]) have analysed challenges and opportunities in defining open datasets for monitoring the diffusion of COVID-19.

Galaitsi and colleagues [24] catalogued the various limitations of data availability, aggregation and interpretation during the COVID-19 pandemic and provided recommendations for improving data management in order to better ensure relevance for decision makers. Ashofteh and Bravo [17] published an analysis of three official multi-source datasets (i.e., WHO, ECDC, Chinese Centre for Disease Control and Prevention, CDC), showing noticeable and increasing measurement errors particularly during the expansion of the pandemic outbreak when more countries contributed data for the official repositories. This study also poses the attention on data comparability and points out the need for better coordination and for harmonized statistical methods. Similarly, Wolkewitz et al. [25] provided an analysis of the most important issues in the adoption of statistical methodologies stating that the majority of researchers still apply simple survival analysis techniques that are not suited for complex hospital data with variables that change over time and that have multiple outcomes. Naqvi [26] presented a COVID-19 European Regional Tracker that collates and homogenizes daily COVID-19 cases provided at the sub-national level by 26 European countries from January 2020. The paper discussed the strengths and weaknesses of each national datasets in terms of machine readability and distribution of data at local level. However, these studies either analysed datasets at the general level or focused the attention on specific indicators such as the number of cases or deaths.

The aim of this work is to analyse the national datasets published by healthcare institutional organizations in six European countries (i.e., Belgium, France, Germany, Italy, Spain and UK) capturing at the country level: (1) how data are organized and distributed to the scientific and non-scientific communities; and (2) which are the main topics considered and the indicators provided to describe them as well as their level of detail.

## 2. Materials and Methods

To identify and review institutional datasets on COVID-19 available at the national level, we firstly analysed the LitCovid [27] portal, a newly established literature database for tracking the latest scientific articles about COVID-19 developed by the National Library of Medicine (NLM) in the United States. LitCovid provides essential bibliographic information such as PubMed ID, title, abstract and journal. As the main objectives of this work were to capture datasets used to analyse the pandemic, the literature analysis focused the attention on the Epidemic Forecasting section, which collects papers aimed at modelling and estimating the trend of COVID-19 spread. Each paper was analysed to capture the source of information related to COVID-19 both collecting the data availability statement and screening the methodological section of the manuscript. Subsequently, the extracted datasets were checked to verify whether data sources were within the list of ECDC (European Centre for Disease Prevention and Control). This is a European agency established in 2004 with the aim of strengthening Europe’s defence against infectious diseases. Among a wide spectrum of activities, ECDC covers surveillance, epidemic intelligence, response, scientific advice, microbiology, preparedness, public health training, international relations, health communication and the publication of the scientific journal *Eurosurveillance* [28]. From the beginning of the pandemic, ECDC has regularly produced maps, graphs and infographics to support the understanding of the dynamics of the pandemic [1], providing the main EU/EEA subnational sources as well as worldwide sources. Moreover, country datasets were checked against sources of information reported in the institutional websites reporting COVID-19 data. In particular, we screened the following institutional websites:Belgium: Sciensano [29];France: Ministère des Solidarités et de la Santé [30];Germany: Robert Koch Institute [31];Italy: Ministero Salute [32];Spain: Ministerio de Sanidad, Consumo y Bienestar Social [33];UK: National Health Service [34].

This double-check of both ECDC and the above-mentioned institutional organization websites has been adopted to verify the completeness of data available.

All the results of these analyses are updated at the end of June 2021.

As previously mentioned, each dataset was subsequently analysed under three main perspectives:Organization and documentation of COVID-19 data reported in the websites to highlight strengths and weaknesses in terms of data reachability and accessibility. This analysis also helps identifying possible additional indicators and topics reported by each country.The level of detail provided to describe each indicator, principally in terms of gender and age ranges. Moreover, data distribution by territorial unit has been analysed by applying the Nomenclature of Territorial Units for Statistics (NUTS) system developed by Eurostat that is based on three main divisions depending on the size of the country. Generally, NUTS-1 refers to state, NUTS-2 identifies regions and NUTS-3 corresponds to provinces. In particular, the attention is focused on the following basic indicators:○The number of daily or cumulative cases as well as the number of diagnostic tests carried out to determine them;○The mortality rate, i.e., the number of patients affected by COVID-19 who died;○The number or percentage of patients that have been hospitalized and/or treated in healthcare structures;○The number of individuals who have been vaccinated. This analysis not only determines the availability of data and the type of analysis that can be performed in each country on the basis of the analysed datasets but also draws the attention to the level of comparability that can be achieved between the six countries taken into consideration. From this perspective, it is important to note that discrepancies across countries and regions may be detected in the computation of these indicators, especially in the mortality rate. In particular, a positive test, a COVID-19 compatible clinical picture or a death certificate mentioning COVID-19 do not always mean that COVID-19 is the underlying (main) cause of death, as it may be a contributory factor. This is a crucial aspect to determine the level of comparability of data in particular in a cross-country perspective.Types of indicators used in each country to monitor the spread of the pandemic. In particular, this analysis intends to capture whether all the perspectives of each topic are covered by the relevant dataset.

## 3. Results

### 3.1. Identification of COVID-19 Institutional Sources

Figure 1 shows the flow diagram that summarizes the results of the LitCovid search in tracking up-to-date published research on COVID-19 and SARS-CoV-2 in the biomedical literature [27].

The LitCovid platform reported more than 130,000 papers published in the field of COVID-19. Among them, more than 1700 were published in the Epidemic Forecasting section, with 338 articles reporting information on at least one of the six countries involved in this analysis. Almost half of these articles (N = 152) were excluded from the analysis as they either did not report data that specifically monitored the spread of COVID-19 or adopted data captured at the local level (e.g., municipality) or at the organizational level (e.g., hospital) or analysed exclusively ad hoc collected data not available on institutional national sites. As shown in Table 1, the majority of the remaining 186 papers used data aggregated at the national level gathered from international repositories (N = 104), while the remaining 82 papers relied on institutional and national sources. Table 1 shows that the majority of the studies pertained to the Italian pandemics (128 documents). Moreover, among the total number of papers, 120 focused the attention on one single country, while the remaining 66 analysed data of two or more than six countries.

Table 2 lists the institutional and national datasets reported in the 82 above-selected papers. For the sake of simplicity and to remove duplicates, we checked and navigated each link to capture the webpage where the files or services producing data were exposed. For instance, if a paper reported that data were gathered from the Italian Ministry of Health webpage [32] or from the Italian institutional dashboard [35], we navigated the website to capture the exact position of the dataset in order to avoid multiple references of the same dataset. As described in the methodology section, all datasets were verified against ECDC data sources as well as against data reported in the institutional websites. Table 2 reports the final list of information sources under investigation in the following sections of this study.

### 3.2. Overview of Individual Countries

In this section, datasets exposed by each country are analysed by highlighting strengths and weaknesses considering, in particular, how the webpages and data are organized and displayed. Moreover, a brief discussion on additional indicators reported by each dataset is reported.

#### 3.2.1. Belgium

Data on the diffusion of COVID-19 are collected and distributed by Sciensano [29], a research and national public health institute of Belgium that operates under the authority of the Federal Minister of Public Health and the Federal Minister of Agriculture. Its core business is scientific research in the fields of public health, animal health and food safety. From the beginning of the epidemic (i.e., 31 of March 2020), Sciensano became the authority for the epidemiological follow-up analysis of the COVID-19 in collaboration with its partners and other healthcare actors [36]. Sciensano provides a set of data on a daily basis in the Epistat website, which also publishes data and displays dashboards on (other) infection diseases (e.g., influenza and Sexually Transmitted Infections), as well as mortality rates. Data on the spread of COVID-19 are updated daily and available in CSV (Comma Separated Values) and JSON (JavaScript Object Notation) formats. Each file reports data on a single indicator distributed by age and gender at province, municipality or regional level. An overall file in Excel format is also available and contains all the collected indicators. The COVID-19 section of the Epistat website provides all measures pertaining to the spread of the virus (i.e., confirmed cases, hospitalizations, mortality and testing procedures) as well as the number of individuals vaccinated. Moreover, additional information is reported to capture hospitalisation and death rates among nursing home residents by week and by region/community. All the variables are clearly described, and definitions, data types and examples are provided in a codebook published in English which is directly downloadable from the home page of the website. Of course, considering that Belgium is a multi-language country, all variables are reported in French and Flemish, with particular attention on the name of municipalities and districts.

Finally, the Sciensiano institute provides a step-by-step procedure to request additional data other than public (open) data reported on the website. However, this procedure is subjected to several restrictions, as clearly reported by the institute. For instance, the scope of data requests is limited and only applies to requests in the context of official (subject to peer review) scientific research, while requests for private and non-research purposes, as well as incomplete and unclear requests, are not admitted.

#### 3.2.2. France

Data monitoring COVID-19 diffusion in France is provided by the different regions and published by the Public France Health System (i.e., Santé Publique France) [46], a public administrative authority created in May 2016 under the supervision of the Ministry of Health. Its main mission is to improve and protect the health of populations around three major perspectives: anticipating, understanding and acting. Data on COVID-19 are published at the official open data portal [37] on the open platform for French public data, steered by the Etalab mission, within the general secretariat for the modernization of public action (SGMAP). The platform allows public services to publish public data and civil society to enrich, modify and interpret them in order to coproduce information of general interest. Among the Health and Social topic, the website reports data related to the supply and consumption of care and the efficiency of the health system, public health, drugs, health establishments, etc. These series are provided in particular by the Ministry of Solidarity and Health, the National Health Insurance Fund, the High Authority for Health, the Technical Agency for Information on Hospitalization and other public establishments under the supervision of local authorities. A specific webpage is dedicated to the main reference datasets concerning COVID-19. The portal is organized into subsections, each one related to a specific topic and described on a specific webpage, such as hospitalizations, testing, mortality and vaccination. Each webpage exposes a large number of files in CSV format, and each one contains a small set of indicators distributed by location (i.e., country, region and department), gender and age groups. As mentioned above, files are exclusively downloadable as CSV, while the system does not provide APIs or additional types of data such as XML or JSON. Considering metadata, each CSV file that contains data is accompanied by a CSV file that contains the list of variables, their description, data types and a relevant example. The description is reported both in French and English. Despite the complexity of accessing the several files available, the French website provides a substantial set of additional variables that can be helpful for integrating COVID-19 data, such as the analytical capacity of diagnostic tests, the epidemic spread of COVID-19 variants (i.e., Alpha formerly called UK Variant; Beta formerly called South Africa Variant; Gamma formerly called Brazil Variant; and Delta formerly called India Variant) and access to the emergency room by gender, age group, region and department. Moreover, the French portal provides an open access text file with the list of individuals who have died from the coronavirus, and it reports the name, family name, place of birth and death, etc. What is important to note is that, as stated in the open data portal, the file of deceased persons being established as part of its public service missions does not contain any information relating to privacy regulations. Therefore, it can be communicated to any person who requests it and published online without concealment, since it does not contain any personal data.

#### 3.2.3. Germany

The main official open-data provider in Germany is the Robert Koch Institute (RKI) [31], a public health institute responsible for monitoring public health with the main tasks of detecting, preventing and treating infectious and non-communicable diseases in Germany. As an upper federal agency subordinated to the Federal Ministry of Health, the institute is also involved in the prevention and tracing of infectious disease outbreaks such as the COVID-19 pandemic, the swine flu pandemic in 2009 and the EHEC O104:H4 outbreak in 2011. In particular, for COVID-19, data on risk assessments, spread of the epidemic and epidemiological studies can be found on a specific webpage of the RKI website [47]. Open data on COVID-19, such as the number of cases diagnosed in Germany divided by Länder over time, can be found in the Datenhub website [38] as well as in the GitHub repository [41], where a copy of the RKI variables is reported. In both sources, the numbers of cases and deaths are reported in a single file providing information aggregated by nation, state (NUTS-2) or province (NUTS-3). Additional files can be accessed, containing data for each federal state, so that researchers that are interested in a specific part of the country do not necessarily have to download the complete file. Both RKI and GitHub websites allow downloading files in CSV format, while RKI also provides information in additional formats such as GDB (Geodatabase), JSON, KML (Keyhole Markup Language), etc. In particular, these file formats are useful for importing in a GIS (Geographical Information System) application in order to quickly display the diffusion of the virus in a geographical map. It is important to note that RKI does not publish data about diagnostic tests performed at the regional and local level.

The German portal reports information about vaccines in another website, and this is also performed for hospitalizations due to COVID-19. In the case of vaccination campaign, information is reported as attached data to a specific dashboard distributed by federal states [39]. By evaluating the hospitalization process, this country shows detailed information only on inpatients at critical care and/or intensive care units (ICUs) for each district (NUTS-3) [40], while no data are provided on COVID-19 hospitalizations. This information is also not available in the ECDC website neither at the regional nor at the national level. However, the periodical RKI report [48] provides the number of individuals hospitalized due to COVID-19 and compares the total number of new cases daily diagnosed, but only at national level.

Despite the lack of information on hospitalization, data about ICU inpatients are enriched by the following data: (1) the total number of patients hospitalized in the relevant period that helps capturing the number of beds occupied by COVID-19 and non-COVID-19 patients; (2) the number of ICU beds available at the state and district levels that allows the determination of the percentage of beds that are still available and the level of hospital saturation; (3) the reference population of each state or district which helps to compute the prevalence of the pathology and to determine the percentage of patients infected and treated in the emergency care.

#### 3.2.4. Italy

Data on the diffusion of COVID-19 in Italy are provided by the Italian Civil Protection Department [49], the national body in Italy that deals with the prediction, prevention and management of emergency events. Data are updated daily, and the time-series can be downloaded by using a GitHub repository organized by regions and provinces depending on the type of variables [42]. The home page provides information about the source of data, the list of variables provided and how they are organized in files and folders. The adoption of this service makes the access to data easier through the adoption of the GitHub REST API, which facilitates calls to the data needed. Moreover, data can be downloaded as CSV or JSON files, each one corresponding to the daily data of each of the 20 Italian regions and autonomous provinces. All the indicators are reported in a single file that provides the number of confirmed cases, deaths, recoveries, hospitalizations, patients treated at home and ICU cases, along with the number of daily tests. As mentioned above, all the variables are reported at the regional level (NUTS-2), with the exception of the number of cumulative cases that are reported at the province level (NUTS-3). Considering the granularity of data, all indicators are reported as aggregated data with no distinction as far as gender or age classes are concerned. Data format and descriptions of each variable are described in a single webpage, where the definition is reported both in Italian and in English. Finally, a pictorial representation of the data flow is displayed in order to report who is involved in the collection, control and distribution of data as well as the time schedule. As highlighted in the schema, data are communicated daily—in aggregated datasets—by every region to the national authorities, producing—according to some authors—a lack of standardization in the collection of data.

Considering the vaccination campaign, information about individuals vaccinated, the number of doses injected and the number of doses supplied is collected, managed and published by the extraordinary commissioner for the management of the COVID-19 emergency. This institution was established by the Italian Presidency of the Council of Ministers for the implementation and coordination of containment and contrast measures relative to the epidemiological emergency COVID-19, which includes the implementation of the strategic plan on COVID-19 vaccination [50]. Data on vaccinations are reported in a specific folder in the same account/page of the GitHub service described above [42], with a similar description of data and metadata. The granularity of data is slightly finer compared to the indicators on COVID-19 diffusion with information on gender, age ranges and type of vaccine. All indicators are updated daily.

Despite the absence of structural indicators useful for capturing the saturation of territorial and hospital systems (e.g., number of ICU beds and professionals), during the pandemic waves, the Italian website provides a CSV file containing all the information about the local and national measures adopted to limit the spread of the virus (e.g., lockdown and curfew measures and school closures). These data are particularly important considering that, in this country, closures and other initiatives were taken at the regional and local level and often without central coordination at the national level.

#### 3.2.5. Spain

Regional COVID-19 Spanish data are collected by the Carlos III Health Institute (Instituto de Salud Carlos III; ISCIII), a Spanish public health research institute legally constituted as a public research agency, which is a sort of quasi-autonomous entity under Spanish law. Data are updated daily and provided through the special section of the institute’s website [43], with geographic information and variables on the evolution of the pandemic. The results presented in this website are obtained from the declaration of COVID-19 cases to the National Epidemiological Surveillance Network (RENAVE). Moreover, variables on the diffusion of the pandemic are also available in the GitHub repository [44], where time series reported by regions (Autonomous Communities, CCAA) can be downloaded in CSV files. Similar to Italy, it is important to highlight that each of the different regions might report case numbers with different criteria [18]. In both sources, data are reported disaggregated by gender and age ranges, with a local distribution in regions and provinces. Data on vaccines can be also gathered in the same GitHub repository.

Additional variables are reported to determine the following: (1) the spread of the virus among the healthcare professionals and (2) the institutionalized individuals vaccinated. Moreover, additional structural indicators are available, such as the number of the available ICUs per region before the pandemic outbreak.

#### 3.2.6. UK

The UK government is collecting data and making them officially available through Public Health England (PHE), i.e., the executive agency of the Department of Health and Social Care in the UK [34]. PHE is responsible for the development of the official UK government website for data and insights on coronavirus (COVID-19) [45]. It provides, in a single website, a set of dashboards summarizing the spread of the virus; the actual situation in terms of deaths, vaccines, hospitalizations and testing; and an interactive map that shows 7 day case rate per 100,000 people at a local authority level. In the same menu, it is also possible to download the source data adopted to define the above-mentioned dashboards on the basis of two alternative approaches: (1) clicking on the ready-to-download files as reported in each single graph or (2) defining a customized set of indicators selecting the area of analysis at the national, regional or local authority levels, with options for the type of indicators and data format including CSV, JSON or XML files. As suggested in the website, the user is required to choose a specific area name in order to filter the data to a specific location. Moreover, the selection of the above options generates a permanent link that may be queried as an API service. This provides an alternative and efficient solution to access data routinely even if, as stated on the coronavirus data website, each user is limited to 10 downloads requests per any 100 seconds due to the large volume of data that may be downloaded through this page, with a maximum rate limit of 100 queries per hour. Moreover, each download may contain up to a maximum number of five metrics. The level of territorial details allows the determination of the daily cases at each of more than 149 different UTLAs (Upper Tier Local Authority in England).

Among the set of indicators reported, the form allows directly downloading the time series of reproducibility index (R with t) at the national level.

### 3.3. Cross-Country Analysis

In this paragraph, the different country datasets are compared by focusing on the main indicators adopted for monitoring the diffusion of the COVID-19 outbreak. In particular, the analysis captures the level of disaggregation of the indicators adopted to describe the following main themes: the number of cases and diagnostic tests, mortality, hospitalizations and home care, as well as vaccine procedures and coverage. This analysis is carried out according to two main perspectives. The first one concerns the level of detail adopted by each country to describe each topic addressed, for example, location (i.e., national, regional and local), gender and age. This aims to identify the types of information available, their different granularity and to verify the feasibility of cross-country comparison. The second perspective focuses on the type of variables to establish whether information can be compared at the international level. For instance, considering the collection of data about diagnostic tests, some countries report the number of tests performed while others count the number of individuals who have been tested. Moreover, similar issues have to be considered as the number of antigenic rapid tests, including only the molecular (PCR) swabs, is analysed differently by each country. Another example refers to the diffusion of the virus. In some countries, no information about the current number of individuals positive with COVID-19 is reported.

Results of the first dataset review are reported in Table 3 where the level of detail in terms of gender, age ranges and territorial unit is reported for each country and for each topic addressed. The last row of the table provides an overall point of view at the country level.

Considering the diffusion of the virus, four countries (Belgium, France, Germany and UK) distributed data by both age ranges and gender. All countries with the exception of Germany distributed the number of daily infections by provinces (NUTS-3). However, in France and Belgium, the data are aggregated at the regional level if the analysis also requires the specification of gender and age ranges. If we consider the diagnostic tests, only two countries (France and Germany) provide the number of positively tested persons by age ranges and gender. Considering the distribution by territorial units, Belgium and Italy collect cases distributed by provinces (NUTS-3), while tests are gathered at a lower level of detail (NUTS-2). The different distribution of data on positive cases and tests performed makes it difficult to capture the real diffusion of the virus because the ratio between infected and tested is needed. Moreover, cross-country comparisons cannot be performed when data are not distributed by age ranges and gender or have different distributions by territorial units.

By examining the information about mortality, Italy is the only country that does not provide specific information on both gender and age groups, limiting the disaggregation of data on a regional analysis. Among the other five countries, Spain and UK provide detailed data captured at the province level (NUTS-3), while Germany and Belgium provide data at the regional level. In this distinction, we should consider that, in Belgium, this level of division is equivalent to NUTS-1 due to the size of country and regions. France dedicates a specific section on the data platform to this topic by providing a registry of deceased persons received by the INSEE (i.e., the National Institute of Statistics Economic Studies). Each file contains detailed information relative to the patients’ death by COVID-19 including date of birth and death, place of birth and death, etc. These nominative files, as stated by the website, are not easily manipulated for statistical calculations and are only updated once a month. They also include deaths occurring abroad.

Considering the hospital monitoring situation with respect to the COVID-19 pandemic, two main variables are collected: the number of patients hospitalized in wards and in the ICU. As reported above, additional indicators may be reported by individual countries such as the access to hospital emergency services (e.g., France) or the treatment of patients in nursing homes (e.g., Belgium). However, in this section, the attention is posed on the two main above-mentioned indicators. As reported in Table 3, all countries with the exception of Germany report the same distribution of data with respect to both ward and ICU hospitalizations.

In particular, France and Spain distribute data by both gender and age ranges; UK limits the disaggregation to age; and Italy, Belgium and Germany provide data with no specific distinction by gender and/or age ranges. Considering the distribution by territorial units, while France, Spain and UK provide data aggregated at province level, data are provided at regional level in the remaining countries. Note that, in France, data distributed at the province level do not provide details on gender and age ranges, while data are reported at the regional level in order to obtain this level of granularity. As mentioned above, Germany provides hospitalization data only on patients treated in the ICU departments, while no information is reported at the ward level. Finally, at the beginning of 2021 all countries started to monitor the implementation of their vaccine programs by providing information on doses injected, individuals partially or fully vaccinated and doses available over the territory. In this case, the aggregation level differs between countries: while Belgium, France and Italy provide data distributed by gender and age ranges, Germany, Spain and UK limit their data distribution by territorial units. The majority of the countries (Belgium, France, Germany, Italy and Spain) provide vaccination rates at the regional level, while UK provides this information at the province level. As reported above, the NUTS classification of the territorial units varies across countries and, even if Belgium provides data at regional level, the distribution is classified at NUTS-1 level.

An additional analysis is shown in Table 4 where a set of possible indicators are provided to capture the different perspectives covered by each country in the description of each topic.

By examining the indicators measuring the diffusion of the virus, information about the daily or cumulative number of cases may be not sufficient as they do not capture what is the current number of infected individuals. This makes it necessary to capture daily actual cases or recoveries. In this perspective, only Germany, Italy and Spain reported these indicators in their dataset, while Belgium, France and UK limit their data collection on daily or cumulative number of cases. As reported above, these data may be analysed in combination with the laboratory tests carried out. In this case, two indicators may be collected: the number of tests performed in a single day and the number of individuals tested. In the first case, the indicator does not consider the individuals tested and whether they have already had a swab in the previous day or whether they were positively tested in previous analyses. Thus, these data measure a different cluster that cannot be confused with the number of individuals tested. Italy is the only country that reports data on both indicators, while France, Germany, Spain and UK focus on the number of individuals tested. Belgium is the only country that collects data on the number of diagnostic tests.

In the previous section, we have already analysed the mortality indicator that reports the total number of deceased patients affected by COVID-19. It is interesting to note that only France and Spain provided specific information on the location where the patients died. In particular, both reported hospital deaths without distinguishing between ICU and ward deaths, while no countries reported detailed information on deaths occurred at home. Of course, this distinction may be included in the datasets as it provides additional information on disease severity of the individuals affected by COVID-19.

Considering the location of treatment, we focus the attention on hospital and home care by trying to capture whether countries provide specific information on the number of individuals treated in each location. The majority of the countries provided information on daily and actual patients hospitalized for COVID-19, with the exception of Spain which limits its data provision on daily cases treated in the hospital. As mentioned above, Germany provides data only on hospitalizations in the ICU. Similarly, Italy reports data for daily cases only in ICU, but it is the only country that provides information on actual patients treated at home.

Finally, vaccination programmes are monitored in all countries with both the number of doses injected and the number of patients partially or fully vaccinated. Germany and Italy are the only two countries that enrich these data with the availability of vaccine dose so as to determine the percentage of doses injected compared to those available. This information is crucial for capturing the efficacy of the vaccination campaign that can be conditioned by the availability of vaccine doses.

The key differences and impacts of data resources defined and reported across multiple countries and shown in Table 3 and Table 4 are tested in the following case study, which analyses the impact of vaccination programmes in the spread of the virus. We consider, in particular, the relationship between vaccination coverage and positivity rate. This case study intends to pose the attention on both the level of disaggregation in terms of gender, age range and territorial levels of each measure and on the differences in the types of measures adopted to describe it. Starting from the positivity rate, the high level of granularity across countries is represented by a NUTS-2 disaggregation at the territorial level, while the positivity rate cannot be compared at both gender and age range levels, which are considered only by two countries (i.e., France and Germany). The same result was found for vaccination coverage with distribution by gender and age ranges taken into account only by three countries (i.e., Belgium, France and Italy). A cross-country analysis is even more difficult considering the non-homogeneity of measures collected: this is particularly evident for the testing procedures that consider both the number of tests and the number of individuals tested only in Italy, thus making it difficult to provide comparable values.

## 4. Discussion

This study provides detailed analysis of the main features of national datasets published by institutional authorities in six European countries: Belgium, France, Germany, Italy, Spain and UK. The starting point of the analysis was the scientific articles on COVID-19 available at the Epidemic Forecasting section of the LitCovid portal that collects studies indexed by PubMed. Each dataset has been analysed under three main perspectives: (1) the organization and documentation of data to highlight strengths and weaknesses in terms of reachability and accessibility of the relevant dataset; (2) the level of detail adopted to aggregate data considering gender, age ranges and territorial level; and (3) the type of indicators used to describe the main features of the pandemic, such as spread of the virus, mortality and vaccination.

Considering reachability and accessibility, each country defined a specific section of the institutional website to describe the type of data exposed. In particular, the majority of them provided data using self-developed databases (UK and Germany) or GitHub API (Germany, Italy and Spain). This simplifies accessibility as well as the findability of datasets thanks to permanent links with which researchers can use to access data routinely. All countries provided access to both data and metadata requiring no authentication or authorization procedures and adopting open data licenses, such as Creative Commons 4.0. Despite small differences, all countries provided data by using downloadable CSV files. Additional file formats are present, such as JSON, XML or Geodatabase to be directly included in GIS applications. Each dataset is generally described by a limited set of information and metadata mainly through a webpage reporting this information in the original language and in English. However, the absence of controlled vocabularies, ontologies, thesauri and data models render the integration of data and the performance of cross-country analyses hard to accomplish. This is evident considering that variables are generally instantiated by using the original language both for the name and the value of the variable. Moreover, the comparability of data is also affected by the absences of detailed descriptions of data flow and provenances that are not sufficiently reported by institutions. This is crucial especially in regional-based healthcare systems, such as in Italy and Spain [18,51,52], where information is transmitted daily by each region to the national authorities, biassing the count of case numbers and tests performed, if data are not based on harmonised standard procedures. Despite these issues, data timeliness represents one of the most important achievements of data quality for all COVID-19 datasets, as information are mainly updated daily and distributed at local and national levels. This feature represents one of the most important step forward in the provision of institutional open data given that, usually, healthcare data are generally available only one or two years after collection by both national and international authorities. A clear example is represented by the vaccination coverage across Europe for children, with data published by WHO and UNICEF at least one year after their collection.

Considering the indicators and variables adopted to monitor the spread of the pandemic among the six analysed countries, France is the one with finer and more coherent data distribution, as it adopts variables related to gender and age groups as well as data distributed by provinces or regions. This makes it easier not only to establish differences at local, gender and age levels but it also allows summing up virus diffusion, hospitalization and vaccination, reporting a comprehensive view of the impact of the virus throughout the land. Nevertheless, it is worth noting that the French portal provides a full list of deceased individuals by COVID-19, and the list is not updated daily; thus, it is difficult to compute for statistical purposes. Conversely, Italy is the only country that does not consider differences between men and women as well as among age ranges. This is a crucial aspect considering that different papers have been published, and they capture and demonstrate the different responses to viruses by gender and its impact in young, adolescent, adult and elderly individuals. Given that, in Italy, vaccination data are distributed by gender and age ranges, additional efforts could be undertaken to extend the analysis of virus diffusion, hospitalization and mortality. This country also possesses shortfalls in the distribution of data by territorial units, as only cumulative cases are reported at the province level. The other countries have their strengths in the provision of data by providing details on gender and age range details with a limited number of indicators and also reporting differences across topics about the distribution by territorial units.

Finally, the types of indicators adopted by each dataset have been studied in order to gather their level of completeness in terms of capturing the different perspectives of virus diffusion, control, hospitalization, mortality and vaccine compliance. Considering the indicators measuring the diffusion of the virus, only half of the countries (Germany, Italy and Spain) currently map the actual situation by providing not only the number of new daily positive cases but also the actual number of patients considering deaths and recoveries. The other three countries do not provide this information, making it difficult to capture the burden of COVID-19 infections. Another important factor that influenced the treatment of patients and the access to health resources is the place of treatment and place of death. It is well-known that, at least in some of the analysed countries, one of the main issues was the percentage of hospital beds located for COVID-19 patients both in the ICU and in other community healthcare services. The adoption and provision of territorial services at the patient’s home have been recognised as an important factor for limiting the diffusion of the virus and for reducing the mortality among patients and professionals [53,54]. In this perspective, while the majority of the countries provide information on daily and actual patients hospitalized for COVID-19, Italy is the only country that provides information on actual patients treated at home. This information is even more crucial when considering mortality. In fact, no countries report detailed information on home care deaths, while only France and Spain provide specific information on hospital mortality without, however, distinguishing between ICU and ward deaths. Detailed distinctions of treatment and locations of death are worth measuring as they may provide a proxy indicator on disease severity of the individuals affected by COVID-19. Considering vaccination processes, all countries provide similar information on both the number of doses injected and the number of patients partially or fully vaccinated. In addition, Germany and Italy are the only two countries that provide information on the availability of doses, capturing the efficacy of the vaccination campaign and the level of hesitancy of individuals at national and local levels.

## 5. Conclusions

The collection and distribution of real-time open data provided by national and international institutions represent one of the most important tasks to be accomplished already during the first phases of an epidemic spread in order to produce the best-decision making outcomes [55]. During COVID-19, the majority of European countries defined and developed platforms to accomplish this task. This surely represents an important step forward especially when compared to previous initiatives. This is true considering both the number of countries defining their platform to share COVID-19 data and the timeliness in which these data are published, which highly impact their use for epidemiological analysis [21]. Moreover, the quality of information that is generated on a daily basis is exceptionally high for most countries with a proliferation of innovative datasets that provide unique information since the start of the pandemic [26]. However, despite the valuable effort put in place by each national authority to provide prompt data, different issues arise by comparing the datasets provided by the six European countries involved in this study. The major gap, well known in the medical informatics community, mainly relates to the lack of standards and data models shared across institutions. This should consider different perspectives, such as the definition of the indicators computed by each country and the related formula adopted to define them, the data elements and metadata generally reported in the original language without the support of thesauri, vocabularies or nomenclatures. As for other open data collected at national level, COVID-19 indicators are aggregated at different levels by considering gender, age ranges and territorial level. Even if the majority of the countries consider these variables, a heterogeneity both between countries and indicators has been detected. Considering the possibility of cross-country analyses, the highest level of disaggregation by territorial level is provided at the province level (NUTS-3). Even if this level of disaggregation can be considered as an important step forward for the analysis of the epidemic spread, higher levels of data distribution may help scientists to better define how the virus spread across the country, making it possible to adopt measures at the local level. Another important perspective concerns the lack of accompanying variables that may be useful for providing a clear picture on the structural resources available at the national and local level. For instance, considering hospitalization procedures, data on the number of beds available in the ICU and in the medical wards or the medical professionals working in a specific region should be updated and provided to the scientific community in order to provide a comprehensive view of the state of the health system and to rapidly and efficiently assign and reallocate appropriate resources [56,57]. Unfortunately, this aspect has been partially considered and only by a limited number of countries. For instance, the number of doses available by each region in Italy and Germany provides an important indicator to capture the compliance of the vaccine strategy both within and between countries.

In conclusion, the timely response of national and international authorities in the collection and distribution of data on COVID-19 represents an important step forward for the open data community particularly when considering other initiatives in place in Europe so far. These daily data flow may be put in place for routinely collecting and publishing data relative to other pathologies, such as cancer or events such as the compliance to a vaccination program for children or to monitor the diffusion of the seasonal flu. Future initiatives and projects should aim to reach semantic interoperability through the adoption of standards that can facilitate cross-country analyses.

## Figures and Tables

**Figure 1 ijerph-18-10496-f001:**
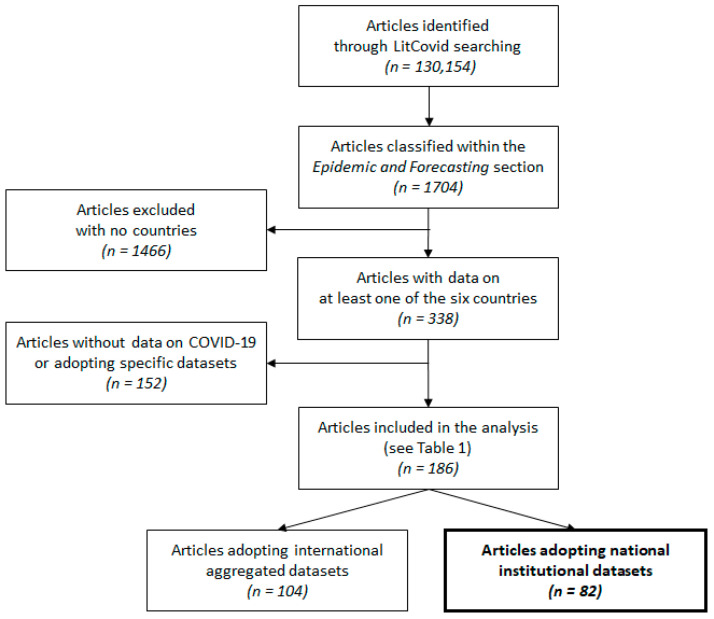
Flow diagram summarizing the LitCovid data retrieval.

**Table 1 ijerph-18-10496-t001:** Distribution of papers by each data source (WHO = World Health Organization; ECDC = European Centre for Disease Prevention and Control; JHU = Johns Hopkins University; Our = Our world in data; WM = world meters).

	WHO	ECDC	JHU	Our	WM	Other	National	Total
Belgium	2	3	1	0	1	0	2	9
France	4	6	13	2	4	1	7	37
Germany	7	5	9	2	6	3	11	43
Italy	18	7	26	2	9	6	60	128
Spain	6	4	12	2	6	1	11	42
UK	1	5	8	2	8	1	7	32
Total *	23	14	36	6	15	10	82	186

* Total specifies the number of papers published adopting a specific dataset considering that each paper may focus its analysis on one or more countries.

**Table 2 ijerph-18-10496-t002:** Source of institutional datasets reported at national level.

Country	Source/Dataset	Dataset Reference
Belgium	https://epistat.wiv-isp.be/Covid/	[36]
France	https://www.data.gouv.fr/fr/pages/donnees-coronavirus	[37]
Germany	Spread monitoring: https://npgeo-corona-npgeo-de.hub.arcgis.com/	[38]
	Vaccine: https://impfdashboard.de/daten	[39]
	Hospitalization: www.arcgis.com/home/item.html?id=8fc79b6cf7054b1b80385bda619f39b8	[40]
	GitHub: https://github.com/jgehrcke/covid-19-germany-gae	[41]
Italy	https://github.com/pcm-dpc/COVID-19	[42]
Spain	https://cnecovid.isciii.es/covid19/	[43]
	https://github.com/datadista/datasets/tree/master/COVID%2019	[44]
UK	https://coronavirus.data.gov.uk/	[45]

Note: All datasets were last accessed on 30 September 2021.

**Table 3 ijerph-18-10496-t003:** Level of detail of each measure at country level considering the gender, age ranges and territorial level.

	Belgium	France	Germany	Italy	Spain	UK
Individuals affected by COVID-19	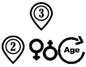	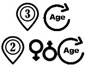	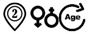	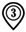	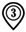	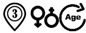
Diagnostic test (all countries report the type of test)	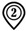	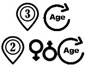	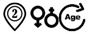	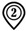	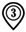	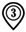
Hospitalization in ward	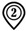	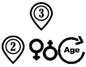	No data available	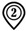	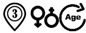	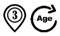
Hospitalization in the Intensive Care Unit	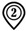	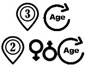	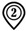	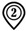	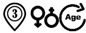	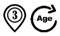
Mortality	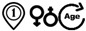	File(s) with data on single patients	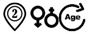	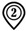	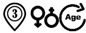	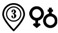
Vaccine (all countries report the type of vaccine)	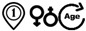	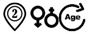	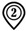	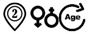	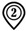	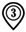
Overview	Partly reported by age and gender. Majority is aggregated at province (NUTS-2) level.	All distributed by age and the majority by gender. All reported at regional (NUTS-2) or departmental (NUTS-3) level.	Part of it is reported by age and gender and aggregated at district level (NUTS-2).	Only vaccination is reported by age and gender. Majority is reported at regional level (NUTS-2).	Part of it is reported by age and gender. Majority is, aggregated at province (NUTS-3) level.	Part of it is reported by age and gender. All indicators are aggregated at local level (NUTS-3).

Legend: level of disaggregation of each measure at the territorial level NUTS-1 (
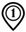
), NUTS-2 (
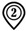
) and NUTS-3 (
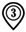
); by gender (
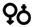
) and by age ranges (
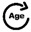
).

**Table 4 ijerph-18-10496-t004:** Level of detail of each topic at country level reporting the availability of the specific set of indicators.

	Belgium	France	Germany	Italy	Spain	UK
**Virus diffusion**
Daily or cumulative cases						
Actual cases						
Recoveries						
**Test**
Diagnostic tests						
Individuals tested						
**Place of treatment**
Daily or cumulative hospitalizations			 (only ICU)	 (only ICU)		
Actual inpatients			 (only ICU)			
Daily or cumulative home care patients						
Actual home care patients						
**Deaths**
Deaths at hospital						
Deaths at home						
**Vaccination**
Individuals vaccinated						
Doses injected						
Availability of vaccines						

## Data Availability

This paper provides an analysis of institutional aggregated datasets reported at the national level that are available at the following websites: (1) Belgium: https://epistat.wiv-isp.be/Covid/; (2) France: https://www.data.gouv.fr/fr/pages/donnees-coronavirus; (3) Germany: (a) https://npgeo-corona-npgeo-de.hub.arcgis.com/; (b) https://impfdashboard.de/daten; (c) www.arcgis.com/home/item.html?id=8fc79b6cf7054b1b80385bda619f39b8&view=list&sortOrder=desc&sortField=defaultFSOrder#overview; (4) Italy: https://github.com/pcm-dpc/COVID-19; (5) Spain: (a) https://cnecovid.isciii.es/covid19/; (b) https://github.com/datadista/datasets/tree/master/COVID%2019; (6) UK: https://coronavirus.data.gov.uk/ (all of the datasets were last accessed on 30 September 2021).

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
