# Peer review of "Open Data Resources on COVID-19 in Six European Countries: Issues and Opportunities"

_ijerph, 2021, doi:10.3390/ijerph181910496_

Round 1

Reviewer 1 Report

In this paper, the open datasets on COVID-19 pandemic from six European countries have been analyzed to understand how they are developed with different indicators and how they are organized and distributed to research and practical communities.  Overall, this work can be valuable to provide guidance on identifying the current data resources and check the availability of different types of information. It can also be useful to compare the difference in data resources across countries. The paper is well written and well organized with sufficient reference and analysis.

One comment for further improvement is to use an example of either research project or policy making practice to illustrate the key differences and impacts of data resources across multiple countries.

For example, in order to research on the forecasting of daily cases, what will be the key impacts of having or missing certain indicators in different data resources?

For cross-country project, what will be the impacts from the difference of data?  Do we need to have any specific adjustments or reconciliations across different data resources?

Author Response

Thank you for the revision. As suggested, we added a short paragraph at the end of the result section to illustrate a case study for a cross-country analysis.

Reviewer 2 Report

The present paper provides a synopsis of publicly accessible datasets about COVID-19 in five European countries. It describes the variables included in the datasets and analyses differences among the countries. Problems of comparability are mentioned. This is sort of useful, but it is not a scientific achievement. Given the limited scientific significance, the text needs to be shortened considerably. This is feasable, because the text is quite redundant. When shortening the paper, the authors need to take care that they use correct English. I recommend professional proofreading. To support the improvement in that direction, I've attached a commented version of the paper with many places highlighted that need correction. (Note that I have not highlighted all language flaws. In particular, I have not marked the many missing commas.) 

The main issues in a little more detail:

  1. I missed a discussion of the problem how death numbers are defined. In some countries, all deaths with a detected COVID-19 infection were counted, regardless of the exact cause of death (e.g. an accident casualty who was infected at the time of death would be counted as COVID-19 death), others defined those numbers more restrictively.
  2. The variables in the dataset should be described more clearly and more systematically. Right now, the descriptions are distributed among the whole text. Moreover, some of the descriptions are not precise enough (see highlighted PDF).
  3. The tables need to be formatted more neatly. In the present state they are hard to read.
  4. Please add a short definition of the NUTS levels.

With respect to the questions about self-citations and plagiarism (see above): I could not find the following reference:

20. Pecoraro F, Luzi D. Beyond the FAIRness of COVID-19 data: what about quality? Stud Health Technol Inform. 2021.

I suppose that paper has not been published as yet. Therefore, I cannot judge how much its content overlaps with that of the present paper. The title gives rise to some suspicion.

Author Response

Thank you for the revision. 

The paper has been updated also shortening the text and removing redundancies. 

All comments and suggestions reported in the PDF file have been considered and included in the manuscript.

A specific paragraph has been added in the methods section to report how countries count the number of death of/affected form COVID-19 

Tables have been updated 

Definition and description of NUTS levels have been included. 

Considering reference 20, it has been accepted by EFMI editorial board and will be published in november in the Studies in health technology and informatics journal with the rest of the EFMI-SPC proceedings. The paper is focused on the same datasets but it analyses their level of FAIRification as well as thier data quality characteristics. It does not overlap with the present paper. I attach the camera ready version for your convenience. 
